

# Polymicrobial detection and salivary metabolomics of children with early childhood caries

Ting Pan, YuJia Ren, JingYi Li, Ying Liao and XiangHui Xing

Department of Pediatric Dentistry, Nanjing Stomatological Hospital, Affiliated Hospital of Medical School, Research Institute of Stomatology, Nanjing University, Nanjing, Jiangsu Province, China

## ABSTRACT

**Background**. Early childhood caries (ECC) has been proposed to be associated with various microorganisms and metabolites. This study aims to compare the prevalence of specific microbial species and salivary metabolomics profile in children with and without ECC, and to explore the correlation between salivary metabolites and targeted microbes.

**Method**. Five ml of unstimulated saliva was collected from 32 ECC and 22 caries-free children. Clinical indexed were recorded and questionnaires regarding oral health and dietary habits were obtained from the guardians. The presence of eight specific microbial species were examined using species-specific quantitative PCR (qPCR). Untargeted metabolomics was analyzed to identify key differential metabolites and pathways. Correlations among clinical, microbial, and metabolomic data were further explored.

**Results**. The prevalence of *Scardovia wiggsiae* (90.6%, $P < 0.001$), *Streptococcus mutans* (43.8%, $P = 0.006$), *Streptococcus sobrinus* (62.5%, $P < 0.001$), *Ligilactobacillus salivarius* (93.6%, $P = 0.01$) and *Candida albicans* (56.3%, $P < 0.001$) were significantly higher in the ECC group. The prevalence of ECC was higher in children with two targeted species present compared with children with one targeted species. Histidine metabolism and branched-chain amino acids degradation were activated in ECC group, while glyoxylate and dicarboxylate metabolism, purine and pyrimidine metabolism were inhibited. Histidine and glutathione metabolism was activated with enrichment of targeted microbial species, while linoleic acid metabolism and biotin metabolism was inhibited. The duration of each toothbrushing was a significant risk factor for ECC experience.

**Conclusion**. The prevalence of *Scardovia wiggsiae, Streptococcus mutans, Streptococcus sobrinus* and *Candida albicans* is higher in ECC children compared to caries-free children. Oral habits and salivary metabolites also vary between ECC and caries-free children.

Corresponding authors
Ying Liao, liaoqiaoling@163.com
XiangHui Xing, dr.xing@nju.edu.cn

## INTRODUCTION

Dental caries remains one of the most common chronic disease of the human oral cavity and can occur at any age. Children with deciduous teeth are more susceptible to caries because

of the risk factors including higher organic content of primary teeth, higher frequency of sugary intake, and relatively poor maintenance of personal oral hygiene. Children under 6 years of age who have one or more caries (cavities or non-cavities), missing (due to caries), or filled surfaces in any deciduous tooth are referred to as early childhood caries (ECC) (*Deng, Zhang & Zou, 2020*). ECC is a multifactorial disease influenced by several factors, including diet, oral hygiene practices, the oral microbiome, and genetic predispositions (*Reisine & Douglass, 1998*; *Park & Choi, 2022*; *De Jesus et al., 2022*; *Kateeb et al., 2023*). Socioeconomic factors such as place of residence and mode of delivery have also been associated with ECC experiences (*Khan et al., 2024*). Infants born vaginally had significantly higher intraoral colonization of *Streptococcus mutans* compared to those born by caesarean section (*Pattanaporn et al., 2013*). A cross-sectional study found that children delivered by caesarean section have a higher caries risk compared to those delivered vaginally (*Felsypremila et al., 2024*). The results of the third National Dental Epidemiologic Survey in China showed that the caries prevalence among 3-year-olds and 5-year-olds reached 34.5% and 71.9%, respectively (*Du et al., 2018*). The etiology of caries has confirmed that microorganisms play an important role in the initiation and progression of caries. Examination of supragingival plaques from preschool-age children revealed that *Streptococcus mutans* (*S. mutans*), *Selenomonas sputigena*, *Pseudomonas salivae*, and *Lactobacillus wadei* were significantly associated with ECC (*Cho et al., 2023*). A large number of previous studies have focused on a classical cariogenic bacterium, *S. mutans* (*Krzyściak et al., 2014*). However, it has been found that *S. mutans* cannot be isolated from some children with ECC (*Tanner et al., 2011*). With the development of microbiology and the diversification of detection methods for oral microorganisms, a variety of other bacteria have been found to be potentially correlated with ECC, including but not limited to *Scardovia wiggsiae* (*S. wiggsiae*), *Streptococcus salivarius* (*S. salivarius*), *Streptococcus sobrinus* (*S. sobrinus*), *Streptococcus parasanguinis* (*S. parasanguinis*), *Ligilactobacillus salivarius* (*L. salivarius*), *Veillonella parvula* (*V. parvula*) and *Candida albicans* (*C. albicans*) (*Tanner et al., 2011*; *Xiao et al., 2018*; *Zhang, Chu & Yu, 2022*). An investigation of these novel caries-associated microorganisms in Chinese ECC children may help explain the high prevalence in the population and contribute to targeted prevention and treatment strategies.

More and more researchers have come to realize the importance of taking host-microbe interaction into consideration when exploring oral infectious diseases (*Lamont, Koo & Hajishengallis, 2018*). Metabolites in saliva, which is the key environment where oral bacteria/fungi meet host cells, have been analyzed in several studies on adult caries. Several biomarkers, including salivary mucins, glycoproteins (sCD14), interleukins (IL-2RA, 4, −13), urease, carbonic anhydrase VI, and urea, has been associated with adult caries (*Antonelli et al., 2024*). A previous study on carbohydrate metabolites of adolescents with caries found significantly different clustering of organic acids in caries-active subjects (*Havsed et al., 2021*). Histamine, L-histidine and succinate emerges are considered to be the major salivary metabolites associated with caries status, reflecting significant alterations in metabolic pathways in caries-active children (*Li et al., 2023b*). In addition, salivary metabolites were significantly altered in children with dental caries studied, including

amino acid metabolism, pyrimidine metabolism, purine metabolism (*Li et al., 2023a*). The application and development of untargeted metabolomic technologies, represented by liquid chromatograph mass spectrometer (LC-MS), allows for detection of metabolites involved in carbohydrate metabolism, amino acid metabolism, lipid metabolism and many other metabolic pathways. A metabolomic study on children with ECC would benefit our understanding of this disease and help identify potential biomarkers for diagnostic and risk prediction purposes.

Thus, the aim of the present study was to explore the prevalence of targeted microbial species and differential salivary metabolites in children with and without ECC in Nanjing District of China. Additionally, the study seeks to explore the correlation between salivary metabolites and targeted microorganisms associated with ECC.

## MATERIALS & METHODS

### Study population and sample collection

The study is a case-control observational study. Subjects were selected from patients who came to the department of pediatric dentistry, Nanjing Stomatological Hospital (Ethical approval: NJSH-2023NL-020-1). A total of 32 children with ECC and 22 caries-free (CF) children were included in this study. All the study participants were required to provide written informed consent from the parent or guardian of the children. Additionally, for children who were developmentally capable of understanding the nature of the study, assent was sought in accordance with ethical guidelines for pediatric research. Inclusion criteria for this study: (1) Aged 6 years or below; (2) no antibiotics were used for 3 months; (3) consent of the parent or guardian to the child's clinical examination and microbiological sampling. Exclusion criteria: (1) children who had systemic disease or congenital disease; (2) children unable to cooperate with sampling or refuse to sample; (3) bacteria or severe infections in other parts of the body. Two examiners with at least 5-years of experience in the department of pediatric dentistry performed dental examinations and DMFT (decay, missing, filling teeth) index according to the World Health Organization (WHO) criteria were recorded (*World Health Organization, 2013*). Examiner reliability was determined using the weighted kappa coefficient based on images of teeth with or without caries. The mean kappa value was 0.73. Subjects were then divided into the CF group (DMFT = 0) and ECC group (DMFT = $13.16 \pm 3.39$). The saliva collection time was between 7:00 and 8:00 AM, with no brushing of teeth on the evening before and the morning of the collection. All participants were provided with a sterile saliva collection tube, and were instructed to spit five mL whole unstimulated saliva. Collected saliva were immediately placed on ice and transferred to laboratory. Saliva was centrifuged at 4 °C for 10 min and the pellets and supernatant were separately stored at −80 °C. The pellets were used for DNA extraction, while the supernatant was used for untargeted metabolomic analysis (*Zhang et al., 2016*).

Meanwhile, the questionnaires were completed by the parents or guardians. The questionnaire used in our study was designed and modified based on the questionnaire used in the national oral health survey (*Du et al., 2018*). The questionnaire from one child in the caries-free group was excluded in later statistical analysis due to poor quality.

**Table 1  Primers used in this study.**

| Species | | Primer sequence |
|---|---|---|
| Universal 16s | F | GGGACTACCAGGGTATCTAAT |
| | R | GGGACTACCAGGGTATCTAAT |
| *Scardovia wiggsiae* | F | GTGGACTTTATGAATAAGC |
| | R | CTACCGTTAAGCAGTAAG |
| *Streptococcus mutans* | F | AGTCGTGTTGGTTCAACGGA |
| | R | TAAACCGGGAGCTTGATCGG |
| *Streptococcus salivarius* | F | CTGCTCTTGTGACAGCCCAT |
| | R | ACGGGAAGCTGATCTTTCGTA |
| *Streptococcus sobrinus* | F | GACCTGTCAGCCGAAGAACGC |
| | R | CCGCAGAGAAGTATCCCGC |
| *Streptococcus parasanguinis* | F | AACAATGCGATYCCAGTATCRAG |
| | R | CTACGACATTAAAGGTACCDCGG |
| *Ligilactobacillus salivarius* | F | CGAAACTTTCTTACACCGAATGC |
| | R | GTCCATTGTGGAAGATTCCC |
| *Veillonella parvula* | F | GAACGTTTGTTGCGTGCTATTTTTGGT |
| | R | TCGTCGCCATTTTCACGGGTAA |
| *Candida albicans* | F | CACCAACTCGACCAGTAGGC |
| | R | CGGGTGGTCTATATTGAGAT |

General and oral status-associated information was collected: (1) general information of the children; (2) mother-related factors: (a) mode of delivery; (b) feeding patterns within 6 months of birth; (c) whether mother has caries experience; (3) oral health habits: (a) age of starting to brush teeth; (b) whether brush teeth everyday; (c) frequency of brushing everyday; (d) duration of each brushing; (e) use of fluoride toothpaste; (4) dietary habits: (a) frequency of sugary drinks or food; (b) frequency of eating/drinking before bedtime.

## DNA extraction and quantitative PCR

Genomic DNA from the saliva was extracted by DNA extraction kits (TIAamp Bacteria DNA Kit, Tiangen, Beijing, China). DNA concentration and purity was examined using Nano-Drop spectrophotometer using A260/A280 (Nano-Drop one/one^c, Thermo Fisher Scientific, Waltham, MA, USA).

The specific primers used for qPCR are detailed in Table 1. Universal 16s rDNA was used as positive control. A 10-μL reaction protocol was set up for this qPCR screening which consisted of 5 μL ChamQ Universal SYBR qPCR Master Mix (Vazyme, Nanjing, China), 0.5 μL forward and reverse primers and 4 μL template DNA, performed in triplicate. Processing was performed using pre-denaturation at 95 °C for 15 min, followed by 50 cycles denaturation (95 °C for 15 s and 60 °C for 30 s). The last step was melting curve analysis at 95 °C for 15 s, 65 °C for 60 s and 95 °C for 15 s (*Huo et al., 2022*).

## Untargeted metabolomics

LC-MS analysis was used to perform untargeted metabolomics on supernatant of saliva which was collected from ECC and CF children. Two mobile phase condition was used to

improve the metabolite coverage, including both positive ion mode and negative ion mode. Detection and identification of metabolites were performed using MS-dial (ver.5.1.230912) by searching against online database (MoNA, GNPS, HMDB and MS-dial database) and in-house database. Untargeted metabolomic analysis was performed at Cosmos Wisdom company (Hangzhou, China). Detailed methodology is described in the Supplemental Information.

## Statistical analysis
### Clinical data analysis
Children with missing data were excluded from the analysis. The normality of the data was tested through the Shapiro–Wilk test. Nonparametric test was used for continuous data of skew distribution. Demographic variables were summarized and descriptive statistics were reported. The Chi-square test was used to analyze variables including gender, mode of delivery, whether the mother has caries experience, age of starting to brush teeth, whether brushing teeth daily, frequency of daily brushing, use of fluoride toothpaste and frequency of sugary food or drink intake. Fisher's exact test was used to analyze variables including feeding patterns within 6 months of birth, duration of brushing and whether intake of foods before bedtime after brushing. Only the variables that showed significant relation to dental caries experience were studied by multicollinearity analysis and multiple logistic regression analysis. All conclusions were drawn considering the level of significance of 5%. Data analysis was performed using Statistical Package for Social Science version 29.0 (SPSS Inc., Armonk, NY, USA).

### Bioinformatics analysis
R software package was employed to test for significant differences between two groups of samples. $T$-test, Variable importance in projection (VIP) obtained from the orthogonal partial least squares-discriminant analysis (OPLS-DA) model (biological replicates $\geq$ 3) and the $P$/FDR (biological replicates $\geq$ 2) or fold change (FC) values from univariate were used together to screen differential metabolites between two groups. Generally, variables meeting both criteria of $P < 0.05$ and VIP $>1.0$ were considered as differential metabolites. The Fisher's exact test was used to find out whether the differentially expressed metabolites had significant enrichment trends in specific functional types. The differential metabolites identified in the previous step were then subjected to Kyoto Encyclopedia of Genes and Genome (KEGG) pathway. Metabolite Set Enrichment Analysis (MSEA) was performed based on all identified metabolites and KEGG pathway databases using the R package corto (version 1.2.4). Based on the MSEA enrichment results, we screened for significantly enriched pathways and differential metabolites in the pathways by setting a $p$ value threshold ($P < 0.05$) and mapped the metabolic regulatory network using the R package graph (version 2.1.0). Spearman's correlation analysis as well as KEGG pathway enrichment analysis was performed to analyze the correlation between delta Cq ($Cq_{species} - Cq_{universal16s}$) of targeted species and metabolites to screen for significantly up-regulated and down-regulated metabolites associated with the targeted microbes.

**Table 2   General and clinical characterization.**

|  | CF ($n = 21$) | ECC ($n = 32$) | P |
|---|---|---|---|
| Gender |  |  |  |
| Female | 11 (52.4%) | 13 (40.6%) | $\chi^2 = 0.707$, d.f.=1 |
| Male | 10 (47.6%) | 19 (59.4) | $p = 0.4$[a] |
| Mean age $\pm$ SD (years) | 4.74 $\pm$ 0.85 | 4.49 $\pm$ 0.78 | $p = 0.753$[b] |
| dmft | 0 | 13.16 $\pm$ 3.39 |  |

**Notes.**

[a] $P$ of Chi-square test.

[b] $t$-test was adopted to evaluate the association between the age with the caries status.

CF, caries-free; ECC, early childhood caries; dmft, decayed-missing-filled deciduous teeth.

# RESULTS

## General and clinical characterization

A total of 53 children were included in the study. General and clinical informational data of the subjects are shown in Table 2. Age and gender were not statistically different between the CF and ECC groups.

## Questionnaire results

Notably, age of starting to brush teeth, whether brush teeth everyday, frequency of brushing, duration of each brushing, use of fluoride toothpaste and frequency of sugary drinks or food were significantly different in CF and ECC groups (Table 3).

Variables that were significantly differerent between two groups were incorporated into the multicollinearity analysis and followed by the multiple logistic regression model. The multicollinearity analysis showed that there were low levels of collinearity among the variables (Table 4, 1<VIF<10). The multiple logistic regression analysis exhibited that the duration of each brushing was a significant risk factor associated with ECC experience (Table 5).

## Detection and co-detection of specific microbial species

The detection rate of specific microbial species was shown in Table 6. The prevalence of *S. wiggsiae* (90.6%, $P < 0.001$), *S. mutans* (43.8%, $P = 0.006$), *S. sobrinus* (62.5%, $P < 0.001$), *L. salivarius* (93.6%, $P = 0.01$) and *C. albicans* (56.3%, $P < 0.001$) were significantly higher in the ECC group when compared to CF.

Co-detection of *S. wiggsiae*, *S. mutans*, *S. sobrinus*, *L.salivarius* and *C. albicans* in pooled and groupwise samples is shown in Fig. 1. All children with presence of the combination of *C. albicans* and *S. mutans* or *C. albicans* and *S. sobrinus* had ECC (Fig. 1A). Prevalence of ECC was higher in children with two targeted species present compared to that in children with only one targeted species present. Co-detection was more prevalent in ECC group compared with CF group (Fig. 1B).

## Metabolomic results

In the present study, a total of 442 metabolites were detected by untargeted metabolomics. Differential metabolites between ECC group and CF group are shown in Fig. 2. A total of 132 differential metabolites were identified, including 97 up-regulated metabolites and 35

**Table 3  Results from the questionnaires.**

| Factors | CF (n = 21) | ECC (n = 32) | P |
|---|---|---|---|
| Mode of delivery | | | $\chi^2 = 0.167$, d.f. = 1 |
| Spontaneous labor | 13(61.9%) | 18(56.3%) | $p = 0.683$[a] |
| Caesarean section | 8(38.1%) | 14(43.8%) | |
| Birth weight (kg) | 3.35(2.85 ~3.6) | 3.45(3.17 ~3.68) | $Z = -0.938$, $p = 0.348$[b] |
| | | | |
| Feeding patterns within 6 months of birth | | | |
| Formula feeding only | 0(0%) | 5(15.6%) | |
| Formula feeding mainly | 0(0%) | 0(0%) | $\chi^2 = 4.085$, d.f. = 3 |
| Formula and breast feeding equal | 7(33.3%) | 8(25.0%) | $p = 0.239$[c] |
| Breast feeding mainly | 1(4.8%) | 3(9.4%) | |
| Breast feeding only | 13(61.9%) | 16(50%) | |
| Whether the mother has caries experience | | | |
| No | 7(33.3%) | 7(21.9%) | $\chi^2 = 3.382$, d.f. = 2 |
| Yes, treated | 9(42.9%) | 15(46.9%) | $p = 0.184$[a] |
| Yes, untreated | 3(14.3%) | 10(31.3%) | |
| Age of starting to brush teeth | | | $\chi^2 = 4.85$, d.f. = 1 |
| ≤1 years old | 13(61.9%) | 10(31.3%) | $p = 0.028$[a] |
| >1 years old | 8(38.1%) | 22(68.8%) | |
| Brushing teeth everyday | | | $\chi^2 = 10.48$, d.f. =1 |
| Yes | 19(90.5%) | 15(46.9%) | $p = 0.01$[a] |
| No | 2(9.5%) | 17(53.1%) | |
| Frequency of brushing | | | $\chi^2 = 7.62$, d.f. = 1 |
| 1 times/day | 5(23.8%) | 20(62.5%) | $p = 0.006$[a] |
| ≥ 2 times/day | 16(76.2%) | 12(37.5%) | |
| Duration of each brushing | | | $\chi^2 = 4.68$, d.f. = 1 |
| <3 min | 13(61.9%) | 28(87.5%) | $p = 0.045$[c] |
| ≥ 3 min | 12(38.1%) | 4(12.5%) | |
| Use of fluoride toothpaste | | | $\chi^2 = 5.11$, d.f. = 1 |
| Yes | 19(90.5%) | 20(62.5%) | $p = 0.024$[a] |
| No | 2(9.5%) | 12(37.5%) | |
| Frequency of sugary drinks or food | | | |
| ≤ 1 times/week | 5(23.8%) | 4(12.5%) | $\chi^2 = 7.63$, d.f. = 2 |
| 2–4 times/week | 11(52.4%) | 8(25.0%) | $p = 0.022$[a] |
| ≥ 1 times/day | 5(23.8%) | 20(62.5%) | |
| Intake of foods before bedtime (after brushing) | | | |
| ≤ 1 times/week | 16(76.2%) | 16(50%) | $\chi^2 = 3.75$, d.f. = 2 |
| 2–4 times/week | 4(19.0%) | 10(31.3%) | $p = 0.141$[c] |
| ≥ 1 times/day | 1(4.8%) | 6(18.8%) | |

**Notes.**
[a]$P$ of Chi-square test.
[b]$P$ of Mann–Whitney $U$ test.
[c]$P$ of Fisher's exact test.
CF, caries-free; ECC, early childhood caries.

**Table 4  Independent variable multicollinearity analysis.**

| Variable | Tolerance | VIF |
|---|---|---|
| Age of starting to brush teeth | 0.793 | 1.261 |
| Brushing teeth everyday | 0.669 | 1.495 |
| Frequency of brushing | 0.951 | 1.051 |
| Duration of each brushing | 0.811 | 1.234 |
| Use of fluoride toothpaste | 0.926 | 1.079 |
| Frequency of sugary drinks or food | 0.725 | 1.379 |

**Notes.**
VIF, variance inflation factor.

**Table 5  Significant factors in the multiple logistic regression analysis.**

| Factors | B | P | Adjusted OR | 95% CI Lower | 95% CI upper |
|---|---|---|---|---|---|
| Age of starting to brush teeth | | | | | |
| ≤1 years old | | | **1.00** | | |
| >1 years old | 1.061 | 0.229 | 2.891 | 0.531 | 16.275 |
| Brushing teeth everyday | | | | | |
| No | | | **1.00** | | |
| Yes | −1.202 | 0.361 | 0.301 | 0.023 | 3.954 |
| Frequency of brushing | | | | | |
| 1 times/day | | | **1.00** | | |
| ≥2 times/day | −1.464 | 0.262 | 0.231 | 0.018 | 2.983 |
| Duration of each brushing | | | | | |
| <3 min | | | **1.00** | | |
| ≥3 min | −2.109 | 0.029[*] | 0.121 | 0.018 | 0.808 |
| Use of fluoride toothpaste | | | | | |
| No | | | **1.00** | | |
| Yes | −0.428 | 0.731 | 0.652 | 0.057 | 7.459 |
| Frequency of sugary drinks or food | | | | | |
| ≤ 1 times/week | | | **1.00** | | |
| 2–4 times/ week | −0.742 | 0.484 | 0.476 | 0.60 | 3.807 |
| ≥1 times/day | 1.777 | 0.118 | 5.912 | 0.637 | 54.916 |

**Notes.**
[*]$P < 0.05$.
adjust OR,  adjust odds ratio; 95% CI,  95% Confidence Interval.
The bolded values indicate variables which were used as references in the logistic regression analysis.

down-regulated metabolites. The top 10 up/down-regulated differential metabolites are shown in Tables 7 and 8 respectively.

KEGG enrichment analysis and MSEA were performed to determine functional pathways and metabolite sets with major differences between two groups. Figure 3 showed results from KEGG enrichment analysis. Nitrogen metabolism was mostly enriched. D-amino acid metabolism and pyrimidine metabolism contains the most numbers of differential metabolites. MSEA revealed that eight metabolite sets were significantly different between

**Table 6  Detection rates of specific microbial species.**

| | CF | | ECC | | P |
|---|---|---|---|---|---|
| | **Positive** | **Negative** | **Positive** | **Negative** | |
| *Scardovia wiggsiae* | 11 (50%) | 11 (50%) | 29 (90.6%) | 3 (9.4%) | $\chi^2$=11.20, d.f. = 1 $p < 0.001$[a] |
| *Streptococcus mutans* | 2 (9.1%) | 20 (90.9%) | 14 (43.8%) | 18 (56.3%) | $\chi^2$=7.51, d.f. = 1 $p = 0.006$[a] |
| *Streptococcus salivarius* | 22 (100%) | | 32(100%) | | |
| *Streptococcus sobrinus* | 3 (13.6%) | 19 (86.4%) | 20 (62.5%) | 12 (38.7%) | $\chi^2$=12.73, d.f. = 1 $p < 0.001$[a] |
| *Streptococcus parasanguinis* | 20 (90.9%) | 2 (9.1%) | 29 (90.6%) | 3 (9.4%) | $p = 1.000$[b] |
| *Ligilactobacillus salivarius* | 14 (63.6%) | 8 (36.4%) | 30 (93.6%) | 2 (6.3%) | $p = 0.010$[b] |
| *Veillonella parvula* | 22 (100%) | | 32 (100%) | | |
| *Candida albicans* | 2 (9.1%) | 20 (90.9%) | 18 (56.3%) | 14 (43.8%) | $\chi^2$=12.43, d.f. = 1 $p < 0.001$[a] |

**Notes.**
[a] $P$ of Chi-square test.
[b] $P$ of Fisher's exact test.
CF, caries-free; ECC, early childhood caries; d.f., degrees of freedom.

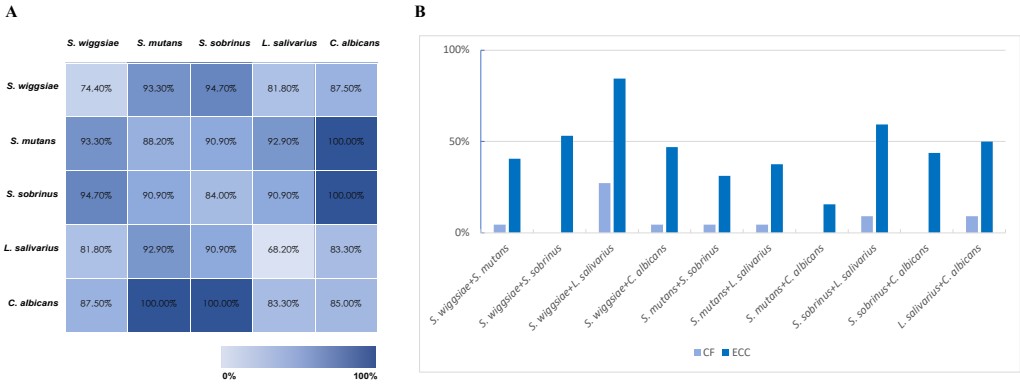

**Figure 1  Co-detection of specific species.** (A) Caries prevalence in pooled samples with co-detection of specific microorganisms. (B) Prevalence of co-existence of specific microorganisms in ECC and CF group.

ECC and CF groups (Fig. 4), including caffeine metabolism, glyoxylate and dicarboxylate metabolism, citrate cycle (TCA cycle), pyrimidine metabolism, histidine metabolism, valine, leucine and isoleucine degradation, pyruvate metabolism and purine metabolism.

Based on the MSEA enrichment results, we screened the pathways with $P < 0.05$ and the different metabolites in the pathway to construct the metabolic regulatory network (Fig. 5). Histidine metabolism and valine, leucine and isoleucine degradation were activated in the ECC group. Conversely, glyoxylate and dicarboxylate metabolism, purine metabolism and pyrimidine metabolism were inhibited in saliva of children with ECC.

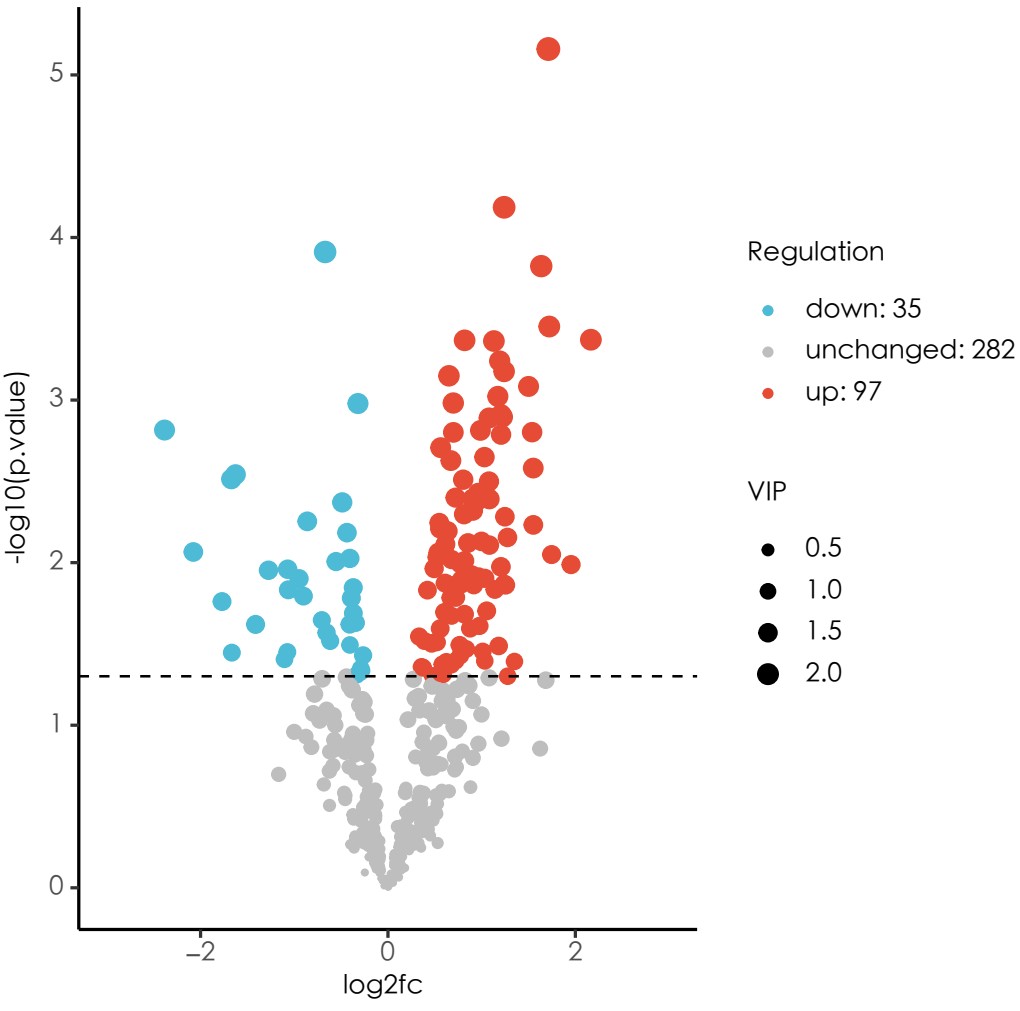

**Figure 2** **Volcano plot of difference analysis.** Blue indicates significantly down-regulated differential metabolites, and red indicated significantly up-regulated differential metabolites, purple-gray indicates metabolites with no significant difference.

## Correlation between microorganisms and metabolomics

Based on the results of the correlation analysis between the Cq (Cq$_{species}$-Cq$_{universal16s}$) values of the targeted specific microbial species and the metabolites, we screened the metabolites with significant correlation with the microbial species content and performed KEGG pathway analysis. Histidine metabolism and glutathione metabolism was activated with enrichment of targeted microbial species, while linoleic acid metabolism and biotin metabolism was inhibited (Fig. 6).

## DISCUSSION

Results from this current study about the detection of multiple acid-producing microorganisms are consistent with the polymicrobial etiology of dental caries suggested by a number of researchers (*Simón-Soro & Mira, 2015*; *Lamont, Koo & Hajishengallis, 2018*;

**Table 7  Detailed information about top 10 up-regulated differential metabolites.**

| Metabolite name | Category | log2fc | *p*- value |
|---|---|---|---|
| Caffeine | Xenobiotics | 3.044292208 | 0.000556028 |
| Paraxanthine | Xenobiotics | 2.165787879 | 0.000425488 |
| Theobromine | Xenobiotics | 1.954194805 | 0.010277702 |
| Arachidonic Acid (C20:4n6) | Lipid | 1.746463203 | 0.00891573 |
| Ser-Ile | Peptide | 1.721707792 | 0.000352883 |
| Cysteine sulfinic Acid | Amino Acid | 1.711880952 | 6.93E−06 |
| 1,3-Diphenylguanidine | Xenobiotics | 1.63612987 | 0.000150028 |
| Histamine | Amino Acid | 1.551612554 | 0.005857378 |
| Spermidine | Amino Acid | 1.550902597 | 0.002619783 |
| Phe-Val | Peptide | 1.538647186 | 0.001575715 |

**Table 8  Detailed information about top 10 down-regulated differential metabolites.**

| Metabolite name | Category | log2fc | *p*- value |
|---|---|---|---|
| Naringenin | Unclassified | −2.384415584 | 0.001528992 |
| Citric Acid | Carbohydrate | −2.077140693 | 0.008597062 |
| Citraconic acid | Unclassified | −1.771183983 | 0.01735429 |
| CDP | Nucleotide | −1.67147619 | 0.003060071 |
| Fructoselysine | Xenobiotics | −1.66630303 | 0.035772992 |
| Aconitic Acid | Carbohydrate | −1.626422078 | 0.002868763 |
| Stachydrine | Xenobiotics | −1.412974026 | 0.023986494 |
| dCMP | Nucleotide | −1.274822511 | 0.011138418 |
| 4-Methylcatechol Sulfate | Xenobiotics | −1.104229437 | 0.03931109 |
| 7-Methylguanosine | Nucleotide | −1.073872294 | 0.035467538 |

*Shao et al., 2023*). The dominant species in early childhood caries seems to differ from that in adolescent and adult caries. While the detection of *Scardovia wiggsiae* has been reported in children with ECC from different populations (*Tanner et al., 2011*; *Simón-Soro & Mira, 2015*; *Chandna et al., 2018*; *Lamont, Koo & Hajishengallis, 2018*; *McDaniel et al., 2021*; *Tantikalchan & Mitrakul, 2022*; *Shao et al., 2023*), the extremely high prevalence of this species in Chinese ECC children (90.6%, Table 6) is firstly reported and suggests that it may play a role in population-specific caries etiology. Notably, *Scardovis wiggsiae* has been found to be more tolerant to fluoride inhibition when compared to *S. mutans* (*Kameda et al., 2020*), indicating a poorer effect of fluoride prevention on dental caries with *Scardovia wiggsiae* being dominant in the oral cavity. While there is no water fluoridation in China, high concentrations of fluoride in groundwater have been reported in different regions of China (*Cao et al., 2022*; *Huang et al., 2023*). It can be hypothesized that the predominance of *Scardovia wiggsiae* could be a result of natural selection. This predominance could also partly explain the relatively compromised protective effect of topical fluoride, which is regularly applied to pre-school children in most areas of China.

Except for *Scardovia wiggsiae*, other microbial species including *S. mutans*, *Streptococcus sobrinus*, *Ligilactobacillus salivarius* and *Candida albicans* were also found to be more

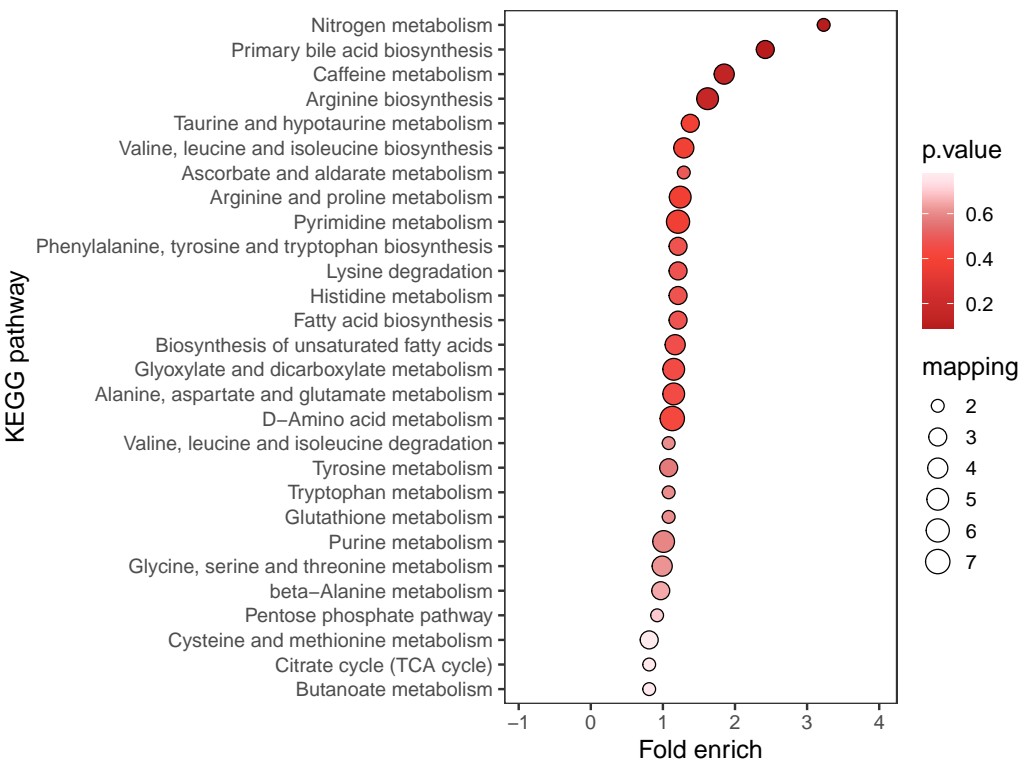

**Figure 3** **KEGG enrichment of ECC *vs* CF.** The results of the top 20 most significantly enriched categories are given in the plot, with the KEGG pathway in the longitudinal axis, the horizontal axis is the Log2-transformed value of the Fold enrichment. Fold enrich is the ratio of GeneRatio to Background Ratio in the corresponding pathway, and a larger value indicates a greater degree of enrichment. The circle colour indicates the *p*-value of enrichment significance, and the circle size indicates the number of different metabolites in the pathway.

prevalent in children with ECC. Among them, *S. mutans* and *S. sobrinus* has been long studied as main pathogens of dental caries and have synergistic effect (*Rupf et al., 2006*; *Li, Wyllie & Jensen, 2021*). High prevalence of *Candida albicans* in ECC has also been reported in numerous studies (*De Carvalho et al., 2006*; *Xiao et al., 2018*; *Sridhar et al., 2020*). The co-existence could greatly promote metabolic activities and biofilm formation of *Candida* and bacteria through direct interaction and indirect signal molecules, thus facilitating caries progression (*Xiao et al., 2018*). In our study, we also noticed a notably higher prevalence of caries when two targeted species were detected in salivary samples, indicating synergistic interactions between different microbes (Fig. 1). Future studies could explore the potential physical/chemical/metabolic interactions and synergies between these targeted species as this may provide deeper insights into their role in caries development. The high detection rate of *Ligilactobacillus salivarius* in children with ECC was somewhat surprising as this species has long been regarded as a probiotic which can inhibit biofilm formation of *S. mutans* and *Candida albicans* (*Krzyściak et al., 2017*; *Wasfi et al., 2018*). Caries incidence and prevalence was reported to be decreased after a short-term intervention with *Ligilactobacillus salivarius* probiotic (*Staszczyk et al., 2022*). However, controversial
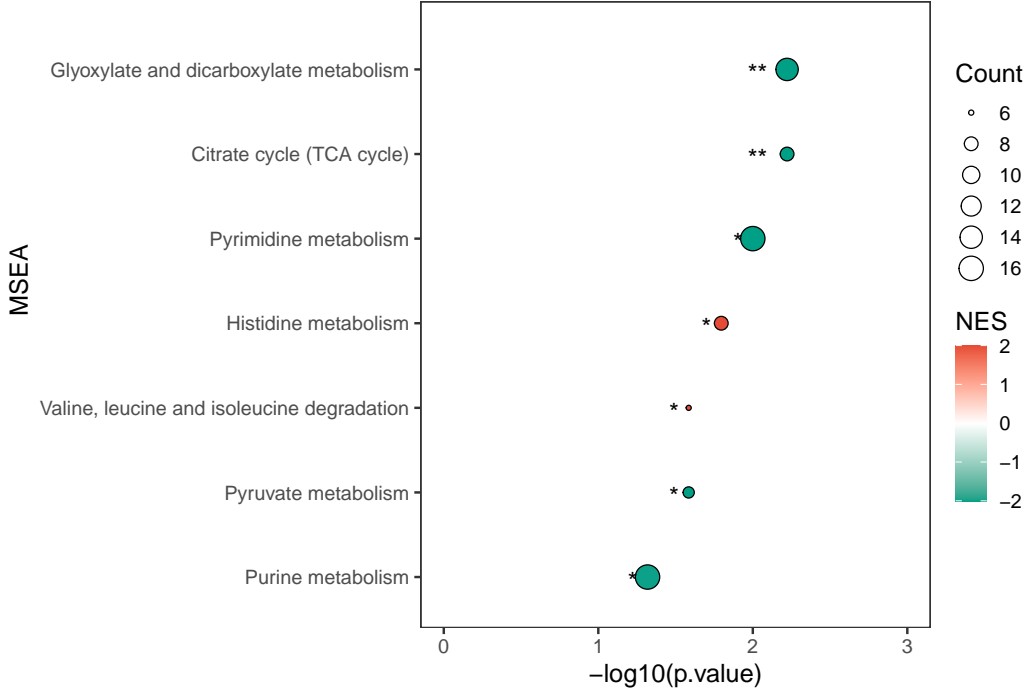

**Figure 4** **MSEA results of ECC *vs* CF.** Normalized enrichment score (NES) was used to measure the enrichment of the metabolite set in the pathway. NES values range from −1 to 1, where greater than 0 indicates that the metabolite set is enriched and less than 0 indicates that the metabolite set is suppressed. Circle size indicates the number of different metabolites in each metabolite set.

results were also reported by other researchers, who found higher level of *Ligilactobacillus salivarius* in adolescents with dental caries (*Shimada et al., 2015*). One explanation for this controversy may due to the genomic diversity of this species and the variety of strains existed in nature (*Neville & O'Toole, 2010*; *Raftis et al., 2011*). More and more researchers have come to realize that virulence of caries-associated bacteria could be strain-specific (*Tanner et al., 2018*; *Al-Hebshi et al., 2019*). A previous study identified two significantly different groups of *Ligilactobacillus salivarius* based on randomly amplified polymorphic DNA fingerprinting (*Švec, Kukletová & Sedláček, 2010*), implicating potentially different metabolic phenotypes. Strain-based study is required to further unveil the cariogenic strains of *Ligilactobacillus salivarius* identified in our current study.

Metabolomic analysis revealed 132 differential metabolites in two groups of children, indicating significantly different salivary status due to microbial activity and host conditions. Amino acid metabolism, especially histidine metabolism, was found to be activated in ECC group (Fig. 5), which is consistent with a previous metabolomic study in children with mixed-dentition (*Li et al., 2023b*). Interestingly, when analyzing metabolomic data based on enrichment of the targeted microbial species, activation of histidine metabolism was also noticed in samples with enriched targeted species (Fig. 6), indicating that microbes selected in the current study can well represent the metabolic status of the ECC oral microcosm. Histidine is an essential amino acid in the human body and plays
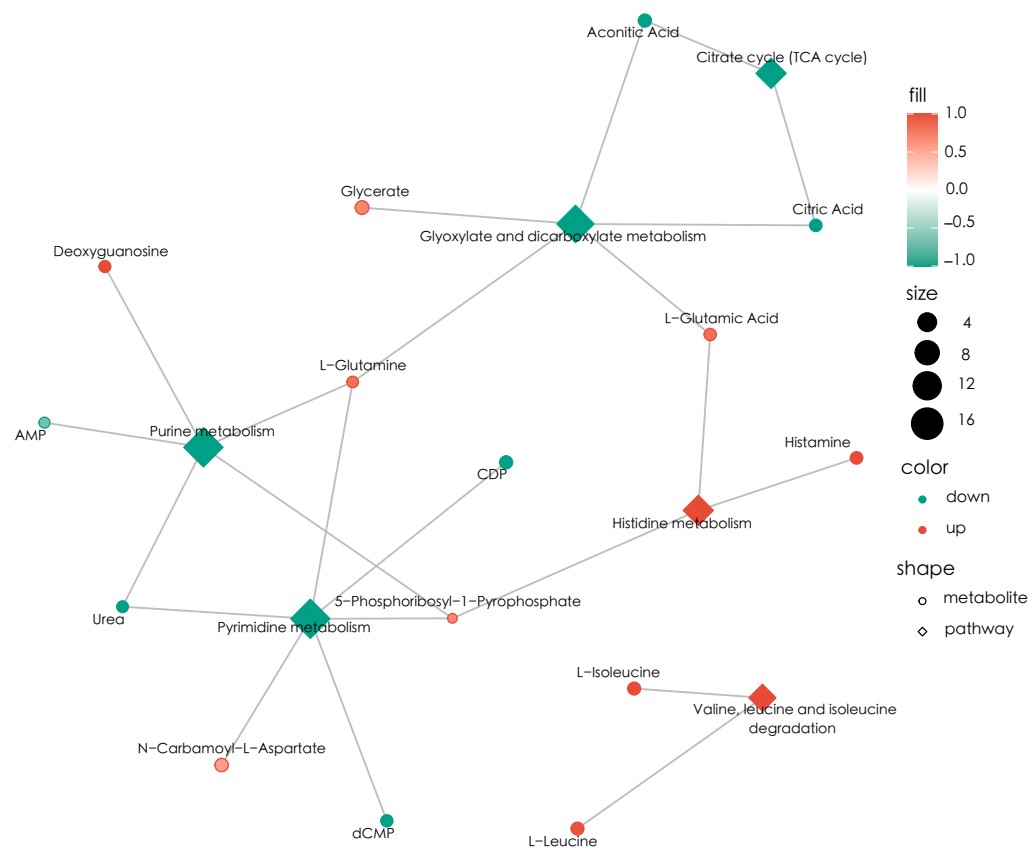

**Figure 5   Differential metabolite regulatory network.** Circles represent differential metabolites, where red indicates up-regulation, green indicates down-regulation, and size indicates VIP values. Diamonds indicate pathways, where red indicates activation and green indicates inhibition, size indicates the number of metabolites detected in the pathway; wiring indicates that metabolites are involved in the pathway process. Connection represents the process in which metabolites participate in this pathway.

a central role in metabolism of histidine-containing proteins (*Moro et al., 2020*). Previous studies have emphasized the significance of salivary histidine-rich proteins, represented by histatin 5, which are important components of the non-specific immune system in the oral cavity (*Jurczak et al., 2015*; *Munther, 2020*). Antimicrobial peptides derived from histatin 5 have been proved to impose killing effect on oral pathogens, including candida and streptococci (*Stewart et al., 2023*; *Skog et al., 2023*). Besides, histatin 5 can inhibit the aggregation of *S. mutans* with other microbiota and thus biofilm formation (*Huo et al., 2011*; *Krzyściak et al., 2015*). In this current study, we detected activated histidine metabolism and higher levels of histamine in ECC group, which is a hydrolytic product of histidine. This result could be associated with lower histatin 5 levels in children with ECC found in other studies (*Jurczak et al., 2015*; *Munther, 2020*). Based on the current result, supplement of histidine, especially histatin 5, should be suggested for children with ECC to improve the host defense against cariogenic microorganisms.

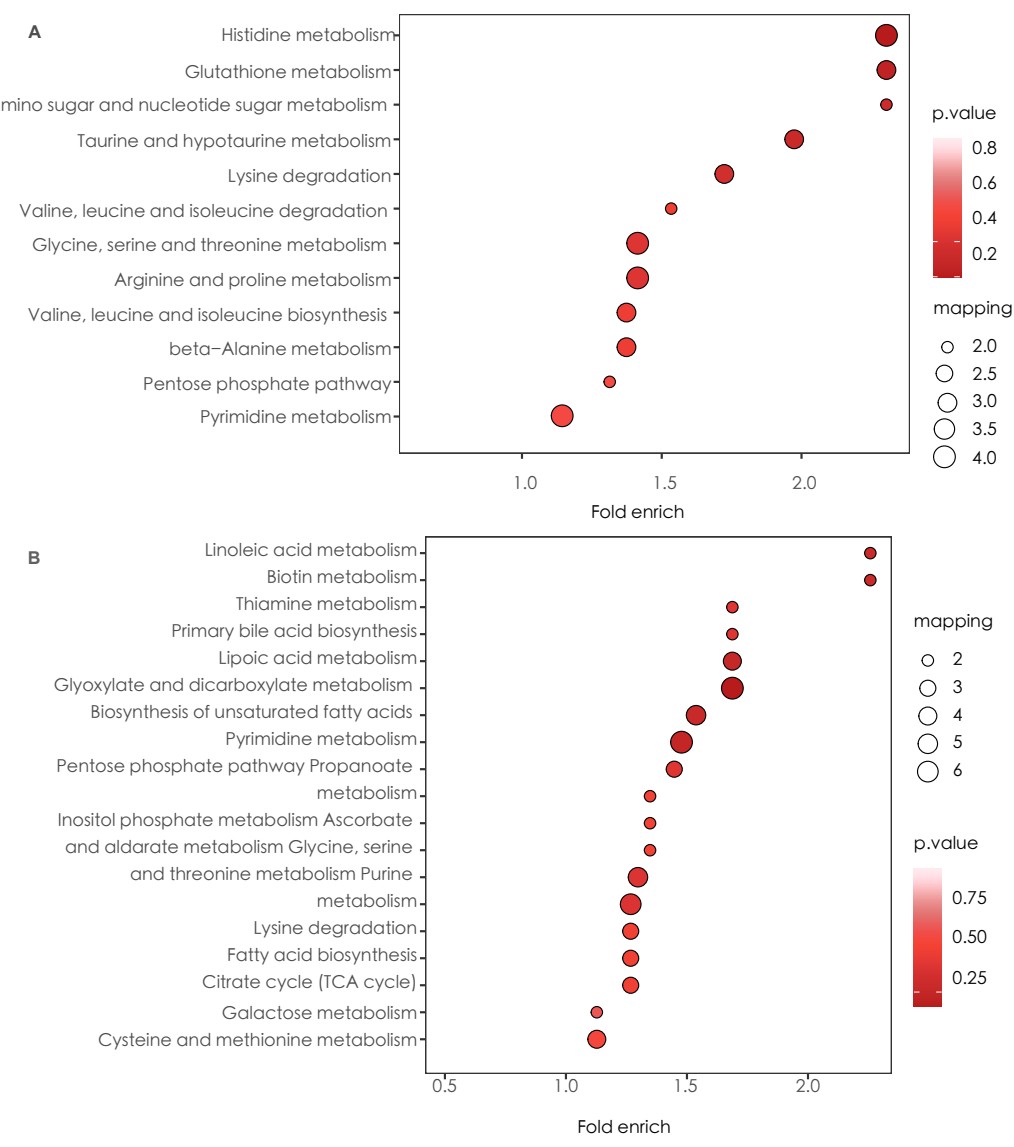

**Figure 6** **KEGG enrichment based on enrichment of targeted microbial species.** (A) Up-regulated metabolic pathways with enrichment of targeted microbial species. (B) Down-regulated metabolic pathways with enrichment of targeted microbial species.

This study also addresses distinct activities of purine and pyrimidine metabolism in children with ECC, correlating with several previous reports which suggested use of purine and pyrimidine-related metabolites as caries biomarkers (*He et al., 2018*; *Li et al., 2023b*). The role of purine and pyrimidine metabolism in caries etiology is still unknown as it has not been found until recent years due to advances in sequencing and metabolomic technologies. Purine and pyrimidine metabolism has been associated with biological interconversion of DNA, RNA, lipids and carbohydrates (*Garavito, Narváez-Ortiz & Zimmermann, 2015*). Pathogens could manipulate pyridine metabolism in host cells to generate an optimal host cellular environment for them to proliferate (*Garavito, Narváez-Ortiz & Zimmermann,*

*2015*). Interestingly, a recent study also reported decreased purine and pyrimidine synthesis in a periodontitis microbial model and the decrease has been hypothesized to be associated with the environmental oxygen gradient caused by bacteria itself (*Yamaguchi-Kuroda et al., 2023*). The recurrent discovery of purine and pyrimidine metabolism change in oral infectious diseases is worth deeper explorations.

The inhibition of glyoxylate and dicarboxylate metabolism in children with ECC is novel in this study. Glyoxylate and dicarboxylate metabolism is well conserved in the entire biosphere and serves as important bypass of the citric acid cycle (TCA) (*Dolan & Welch, 2018*). The central metabolite, glyoxylate, is closely associated with glyoxylate shunt which is involved in oxidative stress, antibiotic stress and host infection in various pathogenic microorganisms (*Xu et al., 2022*). Also, the use of glyoxylate metabolism indicates the ability of the microorganisms to take advantage of carbon sources such as acetate, fatty acids, or ketogenic amino acids (*Dolan & Welch, 2018*). Inhibition of glyoxylate and dicarboxylate metabolism may lead to compromised growth and metabolic potential and competitiveness of the microorganisms in the host environment. Studies about glyoxylate and dicarboxylate metabolism in oral microbes are still rare, suggesting future efforts to understand its role in pathogenesis of caries, especially in ECC.

The current study also included behavioral risk factors of ECC. Factors including age of starting to brush teeth, frequency of brushing, duration of brushing, use of fluoride toothpaste and frequency of sugary intake significantly varied between ECC and CF group, emphasizing the importance of introducing oral health knowledge and habits to parents (*Sun, Zhang & Zhou, 2017*; *Manchanda et al., 2023*). Among the studied factors, brushing teeth for more than 3 min can significantly reduce the risk of caries (Table 5). A very limited number of studies looked into toothbrushing time in children. This issue may reflect the fact that while parents have been told to brush teeth for their children, more specific toothbrushing techniques have not been taught. A previous study on oral health knowledge, attitudes and behavior of parents of preschool children found that parents have different habits and opinions about how to help their children brush their teeth (*Naidu & Nunn, 2020*). In order to enhance the caries preventive effect, more community toothbrushing programs should be addressed to teach parents or guardians to correctly supervise toothbrushing.

This current study locates novel microbial species and salivary metabolites associated with ECC in China, which can be further explored to better understand the disease mechanism and develop targeted preventive agent. Due to the limited sample size, confounding factors, including socioeconomic status and genetic predisposition, were not strictly controlled in this study. Additionally, while we identified significant associations between ECC and microbial/metabolic factors, the cross-sectional design of this study does not allow for causal inference. As the current study did not include direct biofilm measurement, the associations between metabolites such as histatin 5 and cariogenic biofilm formation should further be verified in future *in vitro* studies. Other potential influencing factors, such as immune responses, and environmental exposures, were not extensively analyzed. These limitations should be considered when interpreting the findings, and future

longitudinal studies with larger, more diverse populations and controlled confounders are necessary to validate and further explore these associations.

## CONCLUSIONS

In conclusion, this current study found higher prevalence of *Scardovia wiggsiae* (90.6%), *Streptococcus mutans* (43.8%), *Streptococcus sobrinus* (62.5%), *Ligilactobacillus salivarius* (93.6%) and *Candida albicans* (56.3%) in ECC children compared to caries-free children. Co-detection of two targeted microbial species may indicate higher possibility of caries in children. Oral habits and salivary metabolites also vary between ECC and caries-free children. Metabolomic pathways such as histidine metabolism, purine and pyrimidine metabolism and glyoxylate and dicarboxylate metabolism may reflect host-microbe interaction in ECC status. Key metabolites may be further identified to predict caries risks in children.

### Funding

This work was supported by the Jiangsu Distinguished Medical Expert Fund Project (2020), National Natural Science Funds (82001035) and the ''3456'' Cultivation Program for Junior Talents of Nanjing Stomatological School, Medical School of Nan-jing University (0222R202). The funders had no role in study design, data collection and analysis, decision to publish, or preparation of the manuscript.

### Grant Disclosures

The following grant information was disclosed by the authors:
Jiangsu Distinguished Medical Expert Fund project (2020).
National Natural Science Funds: 82001035.
''3456'' Cultivation Program for Junior Talents of Nanjing Stomatological School.
Medical School of Nan-jing University: 0222R202.

### Competing Interests

The authors declare there are no competing interests.

### Author Contributions

- Ting Pan conceived and designed the experiments, performed the experiments, analyzed the data, prepared figures and/or tables, authored or reviewed drafts of the article, and approved the final draft.
- YuJia Ren performed the experiments, authored or reviewed drafts of the article, and approved the final draft.
- JingYi Li analyzed the data, prepared figures and/or tables, authored or reviewed drafts of the article, and approved the final draft.
- Ying Liao conceived and designed the experiments, authored or reviewed drafts of the article, and approved the final draft.

- XiangHui Xing conceived and designed the experiments, authored or reviewed drafts of the article, and approved the final draft.

## Human Ethics

The following information was supplied relating to ethical approvals (i.e., approving body and any reference numbers):

Nanjing Stomatological Hospital, Medical School of Nanjing University granted ethical approval to carry out the study within its facilities (Ethical Application Ref: NJSH-2023NL-020-1).

## Data Availability

The data is available at Metabolights: MTBLS10890.

## Supplemental Information

Supplemental information for this article can be found online at http://dx.doi.org/10.7717/peerj.19399#supplemental-information.

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
