# Peer review of "Polymicrobial detection and salivary metabolomics of children with early childhood caries"

_PeerJ, doi:10.7717/peerj.19399_

## Round 0.1 · original submission · Major Revisions

Please address comments of all three reviewers and provide a separate response in a point wise manner.

Reviewer 1 ·

Basic reporting

no comment

Experimental design

Dear Editor,

Many thanks for your kind invitation to review this paper, which aimed to compare the prevalence of specific microbial species and salivary metabolomics profile in children with and without early childhood caries and explore the correlation between salivary metabolites and targeted microbes. The idea is interesting. However, the type of study has not been stated, and there are some methodological issues. Other smaller aspects have also been considered below.

Introduction

Lines 53-54: the authors stated that “mode of delivery have also been associated with ECC”, please could you provide the rationality of that statement. As you mentioned in the same paragraph, caries is a multi-factorial oral disease, but there are some factors with stronger evidence and logical explanations of their link with caries,  such as socioeconomic position, oral hygiene, sugar intake, microbiota, etc.

Line 63-64: the statement: “Moreover, topical application of fluoride, a commonly accepted caries-preventive agent, is less effective in preventing caries in children, especially children with high caries risks (Manchanda et al., 2023)”. Please double-check the reference since the statement cannot be based on the Manchanda findings. Moreover, the evidence suggests that some preventive strategies are more effective in people with higher risk.

Lines 71-72: the statement “there have been no reports of prevalence of these novel caries-associated microorganisms in the Chinese ECC community”. That something has not been previously reported or investigated does not justify its conduction. Please do include WHY this is important and how it helps.

The aim of the study is confusing, please provide a concise and unambiguous objective.

The author stated that “the hypothesis was that novel microbial species other than S. mutans would be more prevalent in saliva of ECC children compared to caries-free children” (lines 92-92), but Table 1 shows some microbial species, including the S. mutans and other species that are already associated with caries. Thus, please strengthen the justification of this study.


Method

Please provide the type of study conducted and how the sample size was determined.

Lines 102-103:  the authors stated that “All the study participants were required to provide informed consent from the parent or guardian of the children”, but there is no information about the assent from children.

Lines 104-106: Please review the exclusion criteria, which should not be the opposite of inclusion criteria; instead, they are specific characteristics or conditions that could bias the results.

Line 108: please provide more information about caries diagnosis, which caries index was used? WHO  or ICDAS criteria? Cavitational or pre-cavitational lesions? Moreover, provide data on the inter-examiner agreement for diagnosis.

Please provide references for all methods used in the study, including saliva collection, DNA extraction, qPCR analysis, and untargeted metabolomics. Moreover, include more information in some of them. To illustrate, mention if the saliva was collected after brushing teeth,  at a specific time of the day (morning, afternoon), etc. For PCR analysis, mention how many replicates were used for the sample.

Please provide information about the design and validation of the questionnaires used. Please include information about all variables gathered.

The statistical analysis should be more concise, to illustrate, and mention in which comparisons the authors used chi-square and when used Fischer’s test.

Results

It could be useful to present the age by groups (at least 3 groups)

Table 2, Table 3, and Table 6, please check the statical tests used to compare the caries status by some categorical variables such as gender, mode of delivery, microbial species, etc. Keep in mind the sample size.

Table 5, please provide information about which variables were included in the adjusted models. Also, it is unclear the main outcome.

Double-check the title of the tables.

Figure 1, please provide more details in order to identify the findings by groups.


Discussion

Please include the practical implications of this study. Moreover, acknowledging the limitations and strengths.

Conclusion

Please, double-check the conclusion, which must be based on the main findings of the study. Provide a clear and concise conclusion.

Validity of the findings

no comment

Additional comments

The author should provide the full meaning for all abbreviations or acronyms when they are used for the first time (eg. Line 25 qPCR)

Please update the abstract according to the suggestions made for the main text.

·

Basic reporting

The article is good but needs explanation regarding the methodology

Experimental design

Please explain in detail the number of d (decay) values ​​owned by the subject. The number of teeth experiencing caries in the oral cavity has an impact on the number of microbes in the oral cavity.

Validity of the findings

Has the questionnaire dietary habits used been validated?

·

Basic reporting

The authors present a well-designed study with a reasonable introduction/background to lay the foundation for their work. They also clearly describe the aims of the study and provide sufficient justification for inclusion of chosen variables.

Despite providing a considerable number of references that appropriately cite previous work relating to microbial studies, few references on behavioral drivers of caries are included. A reference should also be cited on line 82 of the Introduction section. It is unclear if the previously referenced Havsed, et al., 2021 study on adolescents included children and if this statement is drawn from that study.

The manuscript uses professional English, but there are some minor grammatical edits throughout that would strengthen the clarity of the writing.

Experimental design

Additional detail in the Methods section regarding the development and content of the questionnaire would strengthen the study and support replication. The authors do not explain how the questionnaire was developed, whether it had been validated, and how it was administered (paper-based, electronic, self-administered, etc.). Limitations associated with the questionnaire should also be noted in the Discussion section.

It is also unclear why 3 minutes was used as the measure of toothbrushing duration. Typically, preventive recommendations suggest 2 minutes so providing an explanation for this specific measure is warranted - especially in light of the significant associations found.

Validity of the findings

The results presented for the questionnaire include a statement on lines 189-190 that draws a conclusion. This statement should be presented in the subsequent Discussion section.

Additional comments

Overall, this is a worthy contribution to the literature that with very minor edits is recommended for publication. Among the minor edits, this reviewer notes several missing spaces between text and citation parentheses throughout the manuscript that should be addressed prior to publication whether by the editors or authors.

---

## Round 0.2 · Minor Revisions

Please address remaining comments of reviewers 2 & 3 and submit revised manuscript along with point wise responces.

Reviewer 1 ·

Basic reporting

The points addressed in my review have satisfactorily been answered and included in the revised paper. Thank you.

Experimental design

The points addressed in my review have satisfactorily been answered and included in the revised paper. Thank you.

Validity of the findings

The points addressed in my review have satisfactorily been answered and included in the revised paper. Thank you.

·

Basic reporting

1. Result
The results of the questionnaire data collection do not explain the results obtained from Table 4. Explain more fully what has been found at this research stage.
2. Discussion
the limitations of the study are less explained more comprehensively

Experimental design

clear

Validity of the findings

conclusion
The statement "higher" should state the value or percentage

Additional comments

revision improved even better

·

Basic reporting

Upon review of the revised manuscript, there are just a few minor grammatical errors that I suggest revising prior to publication to ensure clarity:
Pg. 3, Line 80: replace "understand" with "explain"
Pg. 4, Line 128: should read, "...5 years of experience in the department..."
Pg. 4, Line 132: replace "tooth" with "teeth"
Pg. 7, Line 220: "VIP" should read, "VIF"
Pg. 8, Line 281: should read, "...dominant in the oral cavity." Similarly, on Pg. 9, Lines 322 and 325 add, "the" before oral cavity.
General: When referencing table and figure numbers, remove "the" before the cited table/figure.

Experimental design

No comment

Validity of the findings

Pg. 7, Line 215: Since you previously listed the variables that were significantly different between groups, it is redundant to list them here. I suggest restating this sentence as something similar to what you previously had here, “These variables significantly differed between groups and thus were incorporated into the multiple logistic regression model.”

Discussion section: Given the noted associations between metabolites and specific bacteria with promoting or inhibiting biofilm formation, the authors may want to comment on the lack of biofilm measures in the current study and perhaps recommend inclusion in future studies confirming impacts on biofilm.

Conclusions section: The authors have removed the last line noting the implications of observed metabolomic pathway associations with caries; however, this seems a central finding relevant to the study. I suggest reconsidering its removal and perhaps qualifying the statement with, "may" as in, "...may reflect host-microbe interaction..."

Additional comments

Thank you for your careful consideration of the previously submitted comments. The current version of the article is considerably stronger from the revisions made.

---

## Round 0.3 · Minor Revisions

Please address remaining comments of Reviewer 2 and submit revised manuscript with response to reviewer.

·

Basic reporting

clear

Experimental design

clear

Validity of the findings

clera

Additional comments

Result -> Please state the percentage of each microorganism obtained, according to the statement in the conclusion section.

---

## Round 0.4 · accepted · Accept

Authors have addressed all of the reviewers' comments. I have assessed the revision and I am happy with the current version. This manuscript is ready for publication.